# MULTIGRID NEURAL MEMORY

## ABSTRACT

We introduce a novel architecture that integrates a large addressable memory space into the core functionality of a deep neural network. Our design distributes both memory addressing operations and storage capacity over many network layers. Distinct from strategies that connect neural networks to external memory banks, our approach co-locates memory with computation throughout the network structure. Mirroring recent architectural innovations in convolutional networks, we organize memory into a multiresolution hierarchy, whose internal connectivity enables learning of dynamic information routing strategies and data-dependent read/write operations. This multigrid spatial layout permits parameter-efficient scaling of memory size, allowing us to experiment with memories substantially larger than those in prior work. We demonstrate this capability on synthetic exploration and mapping tasks, where the network is able to self-organize and retain long-term memory for trajectories of thousands of time steps. On tasks decoupled from any notion of spatial geometry, such as sorting or associative recall, our design functions as a truly generic memory and yields results competitive with those of the recently proposed Differentiable Neural Computer (Graves et al., 2016).

## 1 INTRODUCTION

Memory, in the form of generic, high-capacity, long-term storage, is likely to play a critical role in expanding the application domain of neural networks. A trainable neural memory subsystem with such properties could be a transformative technology—pushing neural networks within grasp of tasks traditionally associated with general intelligence and an extended sequence of reasoning steps. Development of architectures for integrating memory units with neural networks spans a good portion of the history of neural networks themselves (*e.g.* from LSTMs (Hochreiter & Schmidhuber, 1997) to the recent Neural Turing Machines (NTMs) (Graves et al., 2014)). Yet, while useful, none has elevated neural networks to be capable of learning from and processing data on size and time scales commensurate with traditional computing systems. Recent successes of deep neural networks, though dramatic, are focused on tasks, such as visual perception or natural language translation, with relatively short latency— *e.g.* hundreds of steps, which is often also the depth of the network itself.

We present a design for neural memory subsystems that differs dramatically from prior architectures in the aspect of read/write interface, and, as a consequence, facilitates parameter-efficient scaling of memory capacity. Our design draws upon both established ideas, as well a key recent advance: multigrid convolutional networks implicitly capable of learning dynamic routing mechanisms and attentional behavior (Ke et al., 2017). LSTMs serve as a fine-grained component, but are wrapped within a larger multigrid connection topology, from which novel capabilities emerge.

In contrast to NTMs and the subsequent Differentiable Neural Computer (DNC) (Graves et al., 2016), we intertwine memory units throughout the interior of a deep network. Memory is a first-class citizen, rather than a separate data store accessed via a special controller. We introduce a new kind of network layer—a multigrid memory layer—and use it as a stackable building block to create deep memory networks. Contrasting with simpler LSTMs, our memory is truly deep; accessing an arbitrary memory location requires passing through several layers. Figure 1 provides a visualization of our approach; we defer the full details to Section 3. There are major benefits to this design strategy, in particular:

- ***Memory scalability.*** Distributing storage over a multigrid hierarchy allows us to instantiate large amounts while remaining parameter-efficient. Read and write operations are similarly distributed. The low-level mechanism underlying these operations is essentially convolution, and we inherit the parameter-sharing efficiencies of convolutional neural networks (CNNs). Our parameterized

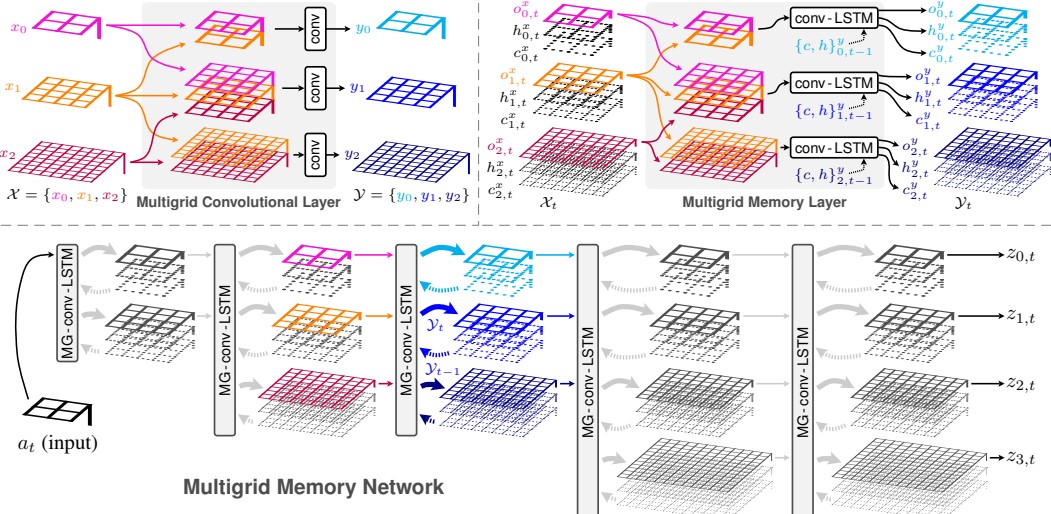

Figure 1: **Multigrid memory architecture.** ***Top Left:*** A multigrid convolutional layer (Ke et al., 2017) transforms input pyramid $\mathcal{X}$, containing activation tensors $\{x_0, x_1, x_2\}$, into output pyramid $\mathcal{Y}$ via learned filter sets that act across the concatenated representations of neighboring spatial scales. ***Top Right:*** We design an analogous variant of the convolutional LSTM (Xingjian et al., 2015), in which $\mathcal{X}$ and $\mathcal{Y}$ are indexed by time and encapsulate LSTM internals, *e.g.* memory cells ($c$), hidden states ($h$), and outputs ($o$). ***Bottom:*** Connecting many such layers, both in sequence and across time, yields a multigrid mesh capable of routing input $a_t$ into a much larger memory space, updating a distributed memory representation, and providing multiple read-out pathways (*e.g.* $z_{0,t}$ or $z_{3,t}$).

filters simply act across a spatially organized collection of memory cells, rather than the spatial extent of an image. Increasing feature channels stored in each spatial cell costs parameters, but adding more cells incurs no such cost, decoupling memory size from the number of parameters.

The multigrid layout of our memory network provides an information routing mechanism that is efficient with respect to overall network depth. We can grow the spatial extent of memory exponentially with depth, while guaranteeing there is a connection pathway between the network input and every memory unit. This allows experimentation with substantially larger memories.

- ***Unification of compute and storage.*** Our memory layers incorporate convolutional and LSTM components. Stacking such layers, we create not only a memory network, but also a generalization of both a CNN and an LSTM. Our memory networks are standard networks with additional capabilities. Though not our primary experimental focus, Section 4 shows they can learn tasks which require performing classification alongside storage and recall.

  A further advantage to this unification is that it opens a wide design space for connecting our memory networks to each other, as well as standard neural network architectures. For example, within a larger system, we could easily plug the internal state of our memory into a standard CNN—essentially granting that CNN read-only memory access. Sections 3 and 4 develop and experimentally validate two such memory interface approaches.

A diverse array of synthetic tasks serves as our experimental testbed. Mapping and localization, an inherently spatial task with relevance to robotics, is one focus. However, we want to avoid only experimenting with tasks naturally fit to the architecturally-induced biases of our memory networks. Therefore, we also train them to perform the same kind of algorithmic tasks used in analyzing the capabilities of NTMs and DNCs. Throughout all experimental settings, DNC accuracy serves as a comparison baseline. We observe significant advantages for multigrid memory, including:

- ***Long-term large-capacity retention.*** On spatial mapping tasks, our architecture retains large, long-term memory. It correctly remembers observations of an external environment collected over paths consisting of thousands of time steps. Visualizing internal memory unit activations actually reveals the representation and algorithmic strategy our network learns in order to solve the problem. The DNC, in contrast, fails to master this category of task.

- ***Generality.*** On tasks decoupled from any notion of spatial geometry, such as associative recall or sorting, our memory networks prove equally as capable as DNCs.

Section 4 further elaborates experimental results, while Section 5 summarizes implications. Before turning to technical details (Section 3), we review the history of coupling neural networks to memory.

## 2 RELATED WORK

There is an extensive history of work that seeks to grant neural networks the ability to read to and write from memory (Das et al., 1992; 1993; Mozer & Das, 1993; Zeng et al., 1994; Hölldobler et al., 1997). Das et al. (1992) propose neural network pushdown automaton, which performs differential push and pop operations on external memory. Schmidhuber (1992) uses two feedforward networks: one that produces context-dependent weights for the second network, whose weights may change quickly and can be used as a form of memory. Schmidhuber (1993) proposes memory addressing in the form of a "self-referential" recurrent neural network that is able to modify its own weights.

Recurrent neural networks with Long Short-Term Memory (LSTMs) (Hochreiter & Schmidhuber, 1997) have enabled significant progress on a variety of sequential prediction tasks, including machine translation (Sutskever et al., 2014), speech recognition (Graves et al., 2013), and image captioning (Donahue et al., 2017). LSTMs are Turing-complete (Siegelmann & Sontag, 1995) and are, in principle, capable of context-dependent storage and retrieval over long time periods (Hermans & Schrauwen, 2013). However, a network's capacity for long-term read-write is sensitive to the training procedure (Collins et al., 2017) and is limited in practice.

Grid LSTMs (Kalchbrenner et al., 2015) arrange LSTM cells in a 2D or 3D grid, placing recurrent memory links along all axes of the grid. This sense of grid differs from our usage of multigrid, as the latter refers to links across a multiscale spatial layout. In Kalchbrenner et al. (2015)'s terminology, our multigrid memory networks are not Grid LSTMs, but are a variant of Stacked LSTMs (Graves et al., 2013).

In an effort to improve the long-term read-write abilities of recurrent neural networks, several modifications have been recently proposed. These include differentiable attention mechanisms (Graves, 2013; Bahdanau et al., 2014; Mnih et al., 2014; Xu et al., 2015) that provide a form of content-based memory addressing, pointer networks (Vinyals et al., 2015) that "point to" rather than blend inputs, and architectures that enforce independence among the neurons within each layer (Li et al., 2018).

A number of methods augment the short- and long-term memory internal to recurrent networks with external "working" memory, in order to realize differentiable programming architectures that can learn to model and execute various programs (Graves et al., 2014; 2016; Weston et al., 2015; Sukhbaatar et al., 2015; Joulin & Mikolov, 2015; Reed & de Freitas, 2015; Grefenstette et al., 2015; Kurach et al., 2015). Unlike our approach, these methods explicitly decouple memory from computation, mimicking a standard computer architecture. A neural controller (analogous to a CPU) interfaces with specialized external memory (*e.g.*, random-access memory or tapes).

The Neural Turing Machine (NTM) augments neural networks with a hand-designed attention-based mechanism to read from and write to external memory in a differentiable fashion. This enables the NTM to learn to perform various algorithmic tasks, including copying, sorting, and associative recall. The Differential Neural Computer (Graves et al., 2016) improves upon the NTM with support for dynamic memory allocation and additional memory addressing modes. Without a sparsifying approximation, DNC runtime grows quadratically with memory due to the need to maintain the temporal link matrix. Our architecture has no such overhead or approximations, nor does it require maintaining an auxiliary state.

Other methods enhance recurrent layers with differentiable forms of a restricted class of memory structures, including stacks, queues, and dequeues (Grefenstette et al., 2015; Joulin & Mikolov, 2015). Gemici et al. (2017) augment structured dynamic models for temporal processes with various external memory architectures (Graves et al., 2014; 2016; Santoro et al., 2016).

Similar memory-explicit architectures have been proposed for deep reinforcement learning (RL) tasks. While deep RL has been applied successfully to several challenging domains (Mnih et al., 2015; Hausknecht & Stone, 2015; Levine et al., 2016), most approaches reason over short-term representations of the state, which limits their ability to deal with partial observability inherent in

many RL tasks. Several methods augment deep RL architectures with external memory to facilitate long-term reasoning. Oh et al. (2016) maintain a fixed number of recent states in memory and then read from the memory using a soft attention-based read operation. Parisotto & Salakhutdinov (2018) propose a specialized write operator, together with a hand-designed 2D memory structure, both specifically designed for navigation in maze-like environments.

Rather than learn when to write to memory (*e.g.*, as done by NTM and DNC), Pritzel et al. (2017) continuously write the experience of an RL agent to a dictionary-like memory module queried in a key-based fashion (permitting large memories). Building on this framework, Fraccaro et al. (2018) augment a generative temporal model with a specialized form of spatial memory that exploits privileged information, including an explicit representation of the agent's position in the environment.

Though we experiment with RL, our memory implementation contrasts with this past work. Our multigrid memory architecture jointly couples computation with memory read and write operations, and learns how to use a generic memory structure rather than one specialized to a particular task.

## 3 MULTIGRID MEMORY ARCHITECTURES

To endow neural networks with long-term memory, we craft an architecture that generalizes modern convolutional and recurrent designs, embedding memory cells within the feed-forward computational flow of a deep network. Convolutional neural networks and LSTMs (specifically, the convolutional LSTM variety (Xingjian et al., 2015)) exist as strict subsets of the full connection set comprising our multigrid memory network. We even encapsulate modern residual networks (He et al., 2016). Though omitted from diagrams for the sake of clarity, in all experiments we utilize residual connections linking the inputs of subsequent layers across the depth (not time) dimension of our memory networks.

Implementing memory addressing behavior is the primary challenge when adopting our design philosophy. If the network structure is uniform, how will it be capable of selectively reading from and writing to only a sparse, input-dependent subset of memory locations?

A common approach is to build an explicit attention mechanism into the network design. Such attention mechanisms, independent of memory, have been hugely influential in natural language processing (Vaswani et al., 2017). NTMs (Graves et al., 2014) and DNCs (Graves et al., 2016) construct a memory addressing mechanism by explicitly computing a soft attention mask over memory locations. This naturally leads to a design reliant on an external memory controller, which produces and then applies that mask when reading from or writing to a separate memory bank.

Ke et al. (2017) recently proposed a multigrid variant of both standard CNNs and residual networks (ResNets). While their primary experiments concern image classification, they also present a striking result on a synthetic image to image transformation task: multigrid CNNs (and multigrid ResNets) are capable of learning to emulate attentional behavior. Their analysis reveals that the network's multigrid connection structure is both essential to and sufficient for enabling this phenomenon.

The underlying cause is that bi-directional connections across a scale-space hierarchy (Figure 1, left) create exponentially shorter signalling pathways between units at different locations on the spatial grid. Information can be efficiently routed from one spatial location to any other location by traversing only a few network layers, flowing up the scale-space hierarchy, and then back down again.

We convert the inherent attentional capacity of multigrid CNNs into an inherent capacity for distributed memory addressing by replacing convolutional subcomponents with convolutional LSTMs (Xingjian et al., 2015). Grid "levels" no longer correspond to operations on a multiresolution image representation, but instead correspond to accessing smaller or larger storage banks within a distributed memory hierarchy. Dynamic routing across scale space (in the multigrid CNN) now corresponds to dynamic routing into different regions of memory, according to a learned strategy. Appendix A provides an analysis of this dynamic routing capability. This routing mechanism is the foundation that enables the addressing capability in multigrid memory architectures.

### 3.1 MULTIGRID MEMORY LAYER

Figure 1 diagrams both the multigrid convolutional layer of Ke et al. (2017) and our corresponding multigrid memory, or MG-conv-LSTM, layer. Activations at a particular depth in our network consist of a pyramid $\mathcal{X}_t = \{(o_{j,t}^x, h_{j,t}^x, c_{j,t}^x)\}$, where $j$ indexes the pyramid level and $t$ indexes time. $o^x, h^x, c^x$ denote the output, hidden state, and memory cell contents of a convolutional LSTM (Xingjian et al.,

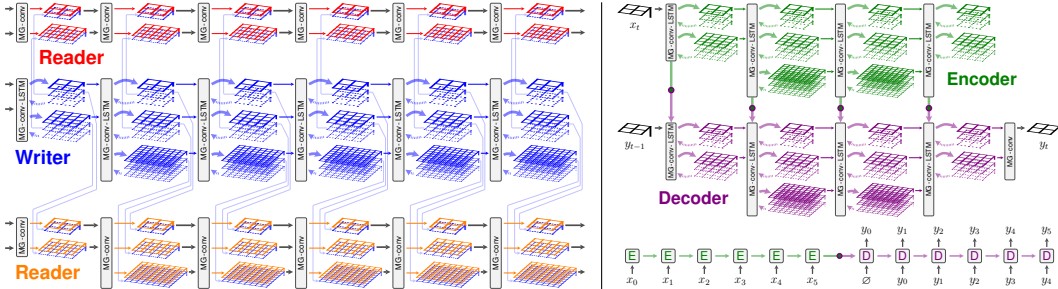

Figure 2: **Memory interfaces.** *Left:* Multiple readers (red, orange) and a single writer (blue) simultaneously manipulate a multigrid memory. Readers are multigrid CNNs; each convolutional layer views the hidden state of the corresponding grid in memory by concatenating it as an additional input. *Right:* Distinct encoder (green) and decoder (purple) networks, each structured as a deep multigrid memory mesh, cooperate to perform a sequence-to-sequence task. We initialize the memory pyramid (LSTM internals) of each decoder layer by copying it from the corresponding encoder layer.

2015), respectively. Following the construction of Ke et al. (2017), outputs $o^x$ at neighboring scales are resized and concatenated, with the resulting tensors fed as inputs to the corresponding scale-specific convolutional LSTM units in the next multigrid layer. The state associated with a conv-LSTM unit at a particular layer and level, say $(o_{j,t}^y, h_{j,t}^y, c_{j,t}^y)$, is computed from memory: $h_{j,t-1}^y$ and $c_{j,t-1}^y$, and input tensor: $\uparrow o_{j-1,t}^x \oplus o_{j,t}^x \oplus \downarrow o_{j+1,t}^x$, where $\uparrow$, $\downarrow$, and $\oplus$ denote upsampling, downsampling, and concatenation, respectively. Specifically, a multigrid memory layer (Figure 1, top right) operates as:

$$O_{j,t}^x := (\uparrow o_{j-1,t}^x) \oplus (o_{j,t}^x) \oplus (\downarrow o_{j+1,t}^x) \tag{1}$$

$$i_{j,t} := \sigma(W_j^{xi} * O_{j,t}^x + W_j^{hi} * h_{j,t-1}^y + W_j^{ci} \circ c_{j,t-1}^y + b^i) \tag{2}$$

$$f_{j,t} := \sigma(W_j^{xf} * O_{j,t}^x + W_j^{hf} * h_{j,t-1}^y + W_j^{cf} \circ c_{j,t-1}^y + b^f) \tag{3}$$

$$c_{j,t}^y := f_{j,t} \circ c_{j,t-1}^y + i_{j,t} \circ \tanh(W_j^{xc} * O_{j,t}^x + W_j^{hc} * h_{j,t-1}^y + b^c) \tag{4}$$

$$o_{j,t}^y := \sigma(W_j^{xo} * O_{j,t}^x + W_j^{ho} * h_{j,t-1}^y + W_j^{co} \circ c_{j,t}^y + b^o) \tag{5}$$

$$h_{j,t}^y := o_{j,t}^y \circ \tanh(c_{j,t}^y) \tag{6}$$

Superscripts denote variable roles (*e.g.* layer $x$ or $y$, and/or a particular parameter subtype for weights or biases). Subscripts index pyramid level $j$ and time $t$, $*$ denotes convolution, $\circ$ Hadamard product. Computation resembles Xingjian et al. (2015), with additional input tensor assembly, and repetition over output pyramid levels $j$. If a particular input pyramid level is not present in the architecture, it is dropped from the concatenation in the first step. Like Ke et al. (2017), downsampling ($\downarrow$) includes max-pooling. We utilize a two-dimensional memory geometry, and change resolution by a factor of 2 in each spatial dimension when moving up or down a pyramid level.

Connecting many such memory layers yields a memory network or distributed memory mesh, as shown in the bottom diagram of Figure 1. Note that a single time increment (from $t - 1$ to $t$) consists of running an entire forward pass of the network, propagating the input signal $a_t$ to the deepest layer $z_t$. Though not drawn here, we also incorporate batch normalization layers and residual connections along grids of corresponding resolution (*i.e.*, from $o_{j,t}^x$ to $o_{j,t}^y$). These details mirror Ke et al. (2017). The convolutional nature of the multigrid memory architecture, together with its routing capability (detailed in Appendix A) provides parameter-efficient addressing of a scalable memory space.

## 3.2 MEMORY INTERFACES

As our multigrid memory networks are multigrid CNNs plus internal memory units, we are able to connect them to other neural network modules as freely and flexibly as one could do with CNNs. Figure 2 diagrams a couple possibilities, which we experimentally explore in Section 4.

On the left, multiple "threads", two readers and one writer, simultaneously access a shared multigrid memory. The memory itself is located within the writer network (blue), which is structured as a deep multigrid convolutional LSTM. The reader networks (red and orange), are merely multigrid CNNs, containing no internal storage, but observing the hidden state of the multigrid memory network.

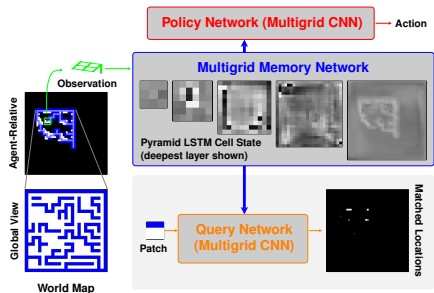 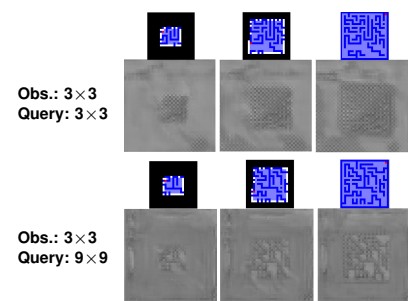

Figure 3: **Mapping, localization, and exploration.** An agent is comprised of a deep multigrid memory, and two deep multigrid CNNs (query and policy subnetworks), which have memory read access. Navigating a maze, the agent makes a local observation at each time step, and chooses a next action, receiving reward for exploring unseen areas. The query subnet, given a local patch, must report all previously observed maze locations matching that patch. Subnet colors reflect Figure 2. *Left:* We view the mean activation, across channels, for LSTM memory cells $\{c_{j,t}\}$ in the network's deepest layer. *Right*: Training only memory and query subnets, using a fixed action sequence (spiral motion), clearly shows the network learns an interpretable strategy for writing and reading memory. Memory contents (hidden states $\{h_t\}$ on deepest, highest-resolution grid) mirror the explored map.

The right side of Figure 2 diagrams a deep multigrid analogue of a standard paired recurrent encoder and decoder. This design substantially expands the amount of addressable memory that can be manipulated when learning common sequence-to-sequence tasks.

## 4 EXPERIMENTS

We first consider a RL-based navigation problem, whereby an agent is tasked with exploring a priori unknown environments with access to only observations of its immediate surroundings. Learning an effective policy requires maintaining a consistent representation of that environment (*i.e.*, a map). Using memory as a form of map, an agent must learn where and when to perform write and read operations as it moves, while retaining the map over long time periods. This task mimics related partially observable spatial navigation scenarios considered by memory-based deep RL frameworks.

*Problem Setup:* The agent navigates an $n \times n$ 2D maze with access to only observations of the $m \times m$ grid ($m \ll n$) centered at the agent's position. It has no knowledge of its absolute position. Actions consist of one-step motion in each of the four cardinal directions. While navigating, we query the network with a randomly chosen, previously seen, $k \times k$ patch ($m \le k \ll n$) and ask it to identify every location matching that patch in the explored map. See Figure 3 (left).

*Multigrid Architecture:* We use a deep multigrid network with multigrid memory and multigrid CNN subcomponents, linked together as outlined in Figure 8. Here, our writer consists of 7 MG-conv-LSTM layers, with maximum pyramid spatial scale progressively increasing from 3×3 to 48×48. The reader, structured similarly, has an output attached to its deepest 48×48 grid, and is tasked with answering localization queries. Figure 3 provides an illustration, while Appendix B.1 provides additional details. Section 4.2 experiments with an additional reader network that predicts actions that drive the agent to explore the maze.

### 4.1 MAPPING & LOCALIZATION

In order to understand the network's ability to maintain a "map" of the environment in memory, we first consider a setting in which the agent executes a pre-defined navigation policy and evaluate its localization performance. We consider different policies (spiraling outwards or a random walk), patch sizes for observation and localization (3×3 or 9×9), as well as different trajectory (path) lengths. We compare against the following baselines:

- Differentiable Neural Computer (DNC) (Graves et al., 2016). See Appendix C for details.
- Ablated MG: a multigrid architecture variant including only the finest pyramid scale at each layer.

Table 1: **Mapping and localization performance.** Our multigrid network for mapping and localization, shown in Figure 3, outperforms the DNC and other baselines by a significant margin. Large memory capacity, enabled by multigrid connectivity, is essential; the DNC even fails to master smaller 15×15 mazes. Our system retains memory over thousands of time-steps. Our localization subnet, trained on random motion, generalizes to answering queries for a *policy-driven agent* (bottom row).

| Architecture | Params (×10⁶) | Memory (×10³) | World Map | FoV | Motion | Query | Path Length | Prec. (%) | Recall (%) | F |
|---|---|---|---|---|---|---|---|---|---|---|
| MG Mem+CNN | 0.65 | 76.97 | | | | | | **99.99** | **99.97** | **99.98** |
| Ablated MG | 1.40 | 265.54 | | | | | | 77.57 | 51.27 | 61.73 |
| ConvLSTM-deep | 0.38 | 626.69 | 25×25 | | | | 529 | 43.42 | 3.52 | 6.51 |
| ConvLSTM-thick | 1.40 | 626.69 | | 3×3 | Spiral | 3×3 | | 47.68 | 1.11 | 2.16 |
| DNC | 0.68 | 8.00 | | | | | | 77.63 | 14.50 | 24.44 |
| DNC | 0.75 | 8.00 | 15×15 | | | | 169 | 91.09 | 87.67 | 89.35 |
| MG Mem+CNN | 0.12 | 7.99 | | | | | | **99.79** | **99.88** | **99.83** |
| | 0.79 | 76.97 | | 3×3 | Spiral | 9×9 | 529 | 97.34 | 99.50 | 98.41 |
| MG Mem+CNN | 0.65 | 76.97 | 25×25 | 3×3 | Random | 3×3 | 500 | 96.83 | 95.59 | 96.20 |
| | | | | | | | 1500 | 96.13 | 91.08 | 93.54 |
| | 0.66 | 78.12 | | 9×9 | Random | 9×9 | 500 | 92.82 | 87.60 | 90.14 |
| | 0.65 | 76.97 | | 3×3 | *Policy* | 3×3 | 1000 | *95.65* | *90.22* | *92.86* |

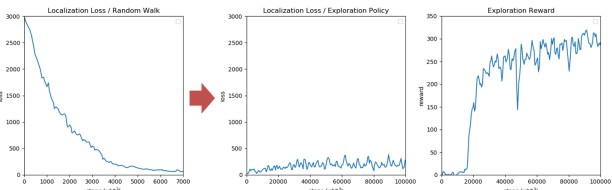

Figure 4: **Generalization of localization.** Fixing parameters after training the query subnet on random motion *(left)*, its localization loss remains low while training the exploration policy *(middle)*, whose reward improves *(right)*.

- ConvLSTM-deep: a deep 23-layer architecture, in which each layer is a convolutional LSTM on a 48×48 grid. This contains the same total number of grids as our 7-layer multigrid network.

- ConvLSTM-thick: 7 layers of convolutional LSTMs acting on 48×48 grids. We set channel counts to the sum of channels distributed across the corresponding layer of our multigrid pyramid.

We train each architecture using RMSProp. We search over learning rates in log scale from $10^{-2}$ to $10^{-4}$, and use $10^{-3}$ for multigrid and ConvLSTM, and $10^{-4}$ for DNC. Randomly generated maps are used for training and testing. Training runs for $8 \times 10^6$ steps with batch size 32. Test set size is 5000 maps. We used a pixel-wise cross-entropy loss over predicted and true locations (see Appendix B.2).

Table 1 reports performance in terms of localization accuracy on the test set. For the simplest setting in which the agent moves in a spiral (*i.e.*, predictable) motion and the observation and query are 3×3, our multigrid architecture achieves nearly perfect precision (99.99%), recall (99.97%), and F-score (99.98%), while all baselines struggle. DNC performs similarly to Ablated MG in terms of precision ($\approx 77.6\%$), at the expense of a significant loss in recall (14.50%). If we instead task the DNC with the simpler task of localization in a 15×15 grid, we see that the performance improves, yet the rates are still around 10% lower than our architecture on the more challenging 25×25 environment. Efficiently addressing large memories is required here, and the DNC temporal memory linkage either incurs quadratic cost for the link matrix, placing a limited cap on the addressable memory ($8k$), or requires a sparsifying approximation. Our architecture has no such overhead or approximations, nor does it require maintaining an auxiliary state. To further evaluate the effect of the reduced memory size, and help putting Multigrid Memory and DNC on an even playground for comparison, we experimented with a reduced version of Multigrid Memory with only $\sim 8k$ memory units, and again found that Multigrid Memory has no issue in mastering this task, with near perfect precision (99.79%), recall (99.88%), and F-score (99.83%).

Figure 3 (right) visualizes the contents of the deepest and high-resolution LSTM block within the multigrid memory network of an agent moving in a spiral pattern. The memory clearly mirrors the contents of the true map, demonstrating that the network has learned a correct, and incidentally, an interpretable, procedure for addressing and writing memory. This further solidifies the addressing capability of the proposed multigrid memory architectures. See Figure 10 for a visualization of the DNC memory, which is not interpretable due to its defined addressing mechanism.

Figure 5: **MNIST recall & classification.** Given a random sequence of images followed by a repeat (green), output the class of the next image (red).

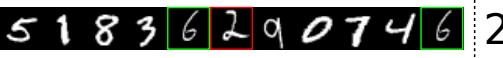

Figure 6: **Learning curves.** Multigrid memory architectures learn significantly faster than others. *Left*: Maze localization task. *Middle*: Joint priority sort and classification. *Right*: Joint associative recall and classification.

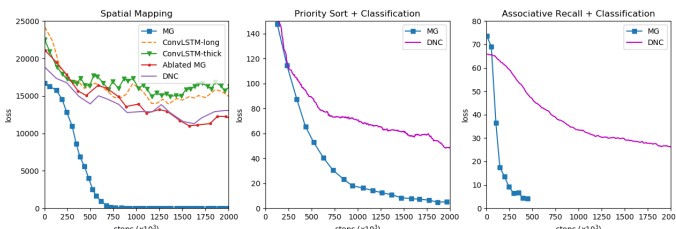

Table 2: **Algorithmic tasks.** Multigrid memory architectures achieve far lower error on the classification variants of priority sort and associative recall, while performing comparably to the DNC on the simpler versions of these tasks. Multigrid memory architectures also remain effective when dealing with long sequences.

| | Architecture | Params $(\times 10^6)$ | Memory $(\times 10^3)$ | Item Size | List Length | Data | Task | Error Rate $\pm \sigma$ |
|---|---|---|---|---|---|---|---|---|
| Standard Sequence | MG Enc+Dec | 0.89 | 76.97 | 3×3 | 20 | Random Patch | Priority Sort | 0.0047±0.0018 |
| | DNC | 0.76 | 8.00 | | | | | 0.0039±0.0016 |
| | MG Enc+Dec | 1.16 | 441.94 | 28×28 | 20 | MNIST | Priority Sort + Classify | **0.0604±0.0081** |
| | DNC | 1.05 | 8.00 | | | | | 0.1659±0.0188 |
| | MG Mem+CNN | 0.65 | 76.97 | 3×3 | 10 | Random Patch | Assoc. Recall | 0.0046±0.0001 |
| | DNC | 0.76 | 8.00 | | | | | 0.0044±0.0001 |
| | MG Mem+CNN | 0.84 | 220.97 | 28×28 | 10 | MNIST | Assoc. Recall + Classify | **0.0312±0.0053** |
| | DNC | 0.90 | 8.00 | | | | | 0.2016±0.0161 |
| Extended Sequence | MG Enc+Dec | 0.89 | 76.97 | 3×3 | 50 | Random Patch | Priority Sort | 0.0067±0.0001 |
| | MG Mem+CNN | 0.65 | 76.97 | | 20 | | Assoc. Recall | 0.0056±0.0003 |

In more complex settings for motion type and query size (Table 1, bottom) our multigrid network remains accurate. It even generalizes to motions different from those on which it trained, including motion dictated by the learned policy that we describe shortly. Notably, even with the very long trajectory of 1500 time steps, our proposed architecture has no issue retaining a large map memory.

## 4.2 Joint Exploration, Mapping, and Localization

We next consider a setting in which the agent learns an exploration policy via reinforcement, on top of a fixed mapping and localization network that has been pre-trained with random walk motion. We implement the policy network as another multigrid reader, and intend to leverage the pre-trained mapping and localization capabilities to learn a more effective policy.

We formulate exploration as a reinforcement learning problem: the agent receives a reward of 1 when visiting a new space cell, $-1$ if it hits a wall, and 0 otherwise. We use a discount factor $\gamma = 0.99$, and the A3C (Mnih et al., 2016) algorithm to train the policy subnet within our multigrid architecture.

Figure 4 (left) depicts the localization loss while pre-training the mapping and localization subnets. Freezing these subnets, but continuing to monitor localization loss, we see that localization remains reliable while a rewarding policy is learned (Figure 4, right). The results demonstrate that the learned multigrid memory and query subnets generalize to trajectories that differ from those in their training dataset, as also conveyed in Table 1 (last row). Meanwhile, the multigrid policy network is able to utilize memory from the mapping subnet in order to learn an effective exploration policy. See the video demo linked in Appendix E for a visualization of learned exploratory behavior.

## 4.3 Algorithmic Tasks

We test the task-agnostic nature of our multigrid memory architecture by evaluating on a series of algorithmic tasks, closely inspired by those appearing in the original NTM work (Graves et al., 2014). For each of the following tasks, we consider two variants, increasing in level of difficulty. See Appendix B.3 and B.4 for complete details.

**Priority Sort.** In the first variant, the network receives a sequence of twenty $3{\times}3$ patches, along with their priority. The task is to output the sequence of patches in the order of priority. Training and testing use randomly generated data. Training takes $2 \times 10^6$ steps, batch size 32, and testing uses 5000 sequences. We tune hyper-parameters as done for the mapping task. We structure our model as an encoder-decoder architecture (Figure 2, right). As revealed in Table 2, our network performs comparably with DNC, with both architectures achieving near-perfect performance.

The second variant extends the priority sort to require recognition capability. The input is a sequence of twenty $28{\times}28$ MNIST images (Lecun et al., 1998). The goal is to output the *class* of the input images in increasing order. Table 2 reveals that our architecture achieves much lower error rate compared to DNC on this task (priority sort + classification), while also learning faster (Figure 6).

**Associative Recall.** In the first task formulation, the network receives a sequence of ten $3{\times}3$ random patches, followed by a second instance of one of the first nine patches. The task is to output the patch that immediately followed the query in the input sequence. We demonstrate this capability using the multigrid reader/writer architecture (Figure 2, left). Training details are similar to the sorting task. Table 2 shows that both DNC and our architecture achieve near-zero error rate.

In the second variant, the input is a sequence of ten $28 \times 28$ randomly chosen MNIST images (Lecun et al., 1998), where the network needs to output the *class* of the image immediately following the query (Figure 5). As shown in Table 2 and Figure 6, our multigrid memory network performs this task with significantly greater accuracy than the DNC, and also learns in fewer training steps.

To further test the ability of multigrid memory architectures in dealing with longer sequences, we experimented the sorting and associative recall tasks with sequence length of 50 and 20, respectively. As can be seen in Table 2, multigrid memory architectures remain effective with near-zero error rates.

The harder variants of both priority sort and associative recall require a combination of memory and pattern recognition capability. The success of multigrid memory networks (and notable poor performance of DNCs), demonstrates that they are a unique architectural innovation. They are capable of learning to simultaneously perform representational transformations and utilize a large distributed memory store. Furthermore, as Figure 6 shows, across all difficult tasks, including mapping and localization, multigrid memory networks train substantially faster and achieve substantially lower loss than all competing methods.

## 5 Conclusion

Our multigrid memory architecture represents a new paradigm in the history of designs linking deep neural networks to long-term storage. We gain dramatic flexibility and new capabilities by co-locating memory with computation throughout the network structure. The technical insight driving this design is an identification of attentional capacity with addressing mechanisms, and the recognition that both can be supported by endowing the network with the right structural connectivity and components: multigrid links across a spatial hierarchy. Memory management is thereby implicit, being distributed across the network and learned in an end-to-end manner. Multigrid architectures efficiently address large amounts of storage, and are more accurate than competing approaches across a diverse range of memory-intensive tasks.

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

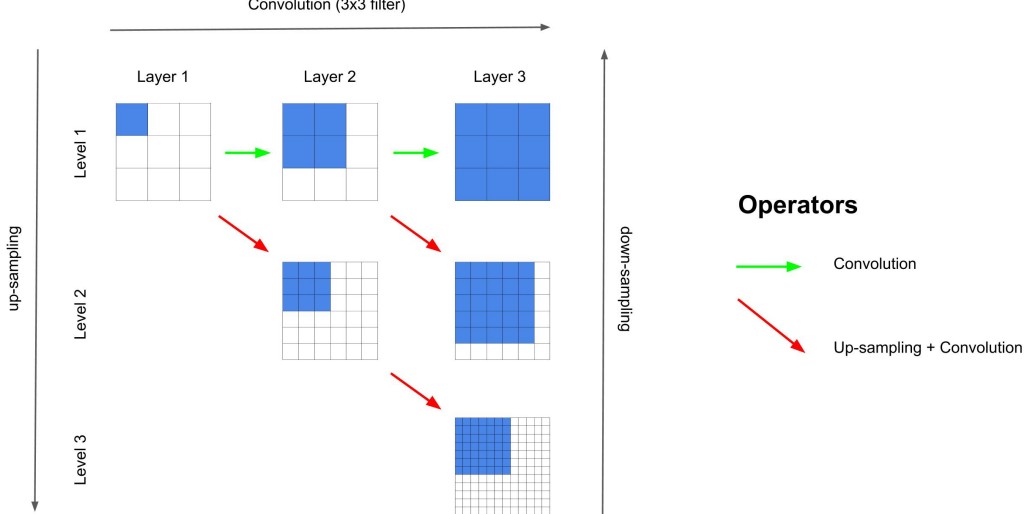

Figure 7: **Information routing.** Shown are example paths of information flow in a multigrid architecture. Progressing from one layer to the next, information flows between grids at the same level (via convolution, green), as well as to adjacent grids at higher resolution (via up-sampling and convolution, red) or lower resolution (via down-sampling and convolution, not shown). Information from location (1,1) (blue) of the source grid at [layer 1, level 1] can be propagated to all the locations marked as blue in subsequent layers and levels, following the indicated paths. With two more layers (not shown), it could reach any location in any level. Similarly rapid routing occurs along down-sampling links (in combination with up-sampling); information on fine-scale levels can quickly flow to coarser levels and then to any location. In a real multigrid network, the portion of this routing capacity actually used is determined by the learned network parameters (convolutional filters).

## A    INFORMATION ROUTING

*Proposition 1:* For the setup in Figure 7, suppose that the convolutional kernel size is $3 \times 3$, and up-sampling is $2\times$ nearest-neighbor sampling. Consider location $(1, 1)$ of the source grid at [layer 1, level 1]. For a target grid at [layer $m$, level $n$], where $m \geq n$, the information from the source location can be routed to any locations $(i, j)$ where $1 \leq i, j \leq (m - n + 2) \cdot 2^{n-1} - 1$.

*Proof:* Induction proof on level $n$.

- For level $n = 1$: Each convolution of size $3 \times 3$ can direct information from a location $(i, j)$ at layer $k$ to any of its immediate neighbors $(i', j')$ where $i - 1 \leq i' \leq i + 1$, $j - 1 \leq j' \leq j + 1$ in layer $k + 1$. Therefore, convolutional operations can direct information from location $(1, 1)$ in layer 1 to any locations $(i', j')$ in layer $k = m$ where $1 \leq i'$, $j' \leq m = (m - 1 + 2) \cdot 2^0 - 1 = (m - n + 2) \cdot 2^{n-1} - 1$.

- Assume the proposition is true for level $n$ ($\forall m \geq n$), we show that it is true for level $n + 1$. Consider any layer $m + 1$ in level $n + 1$, where $m + 1 \geq n + 1$:

  We have, $m + 1 \geq n + 1 \Rightarrow m \geq n$. Therefore, we have that at [layer $m$, level $n$], the information from the source location can be routed to any locations $(i, j)$ where $1 \leq i, j \leq (m - n + 2) \cdot 2^{n-1} - 1$. Now consider the path from [layer $m$, level $n$] to [layer $m + 1$, level $n + 1$]. This path involves the upsampling followed by a convolution operator, as illustrated in Figure 7.

  Nearest-neighbor up-sampling directly transfers information from index $i$ to $2 \cdot i$ and $2 \cdot i - 1$, and $j$ to $2 \cdot j$ and $2 \cdot j - 1$ by definition. For simplicity, first consider index $i$ separately. By transferring to $2 \cdot i$, information from location $1 \leq i \leq (m - n + 2) \cdot 2^{n-1} - 1$ in level $n$ will be transferred to all even indices in $[2, ((m - n + 2) \cdot 2^{n-1} - 1) \cdot 2]$ at level $n + 1$. By transferring to $2 \cdot i - 1$, information from location $1 \leq i \leq (m - n + 2) \cdot 2^{n-1} - 1$ in level $n$ will be transferred

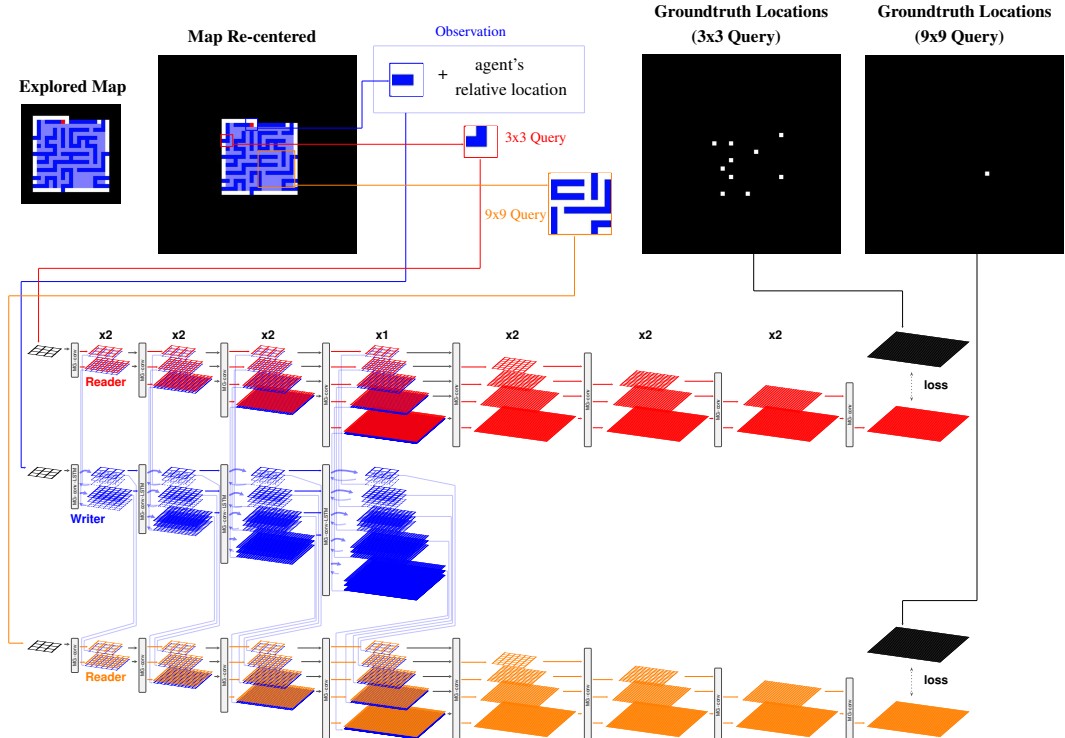

Figure 8: **Multigrid memory writer-reader(s) architecture for spatial navigation.** At each time step, the agent moves to a new location and observes the surrounding $3 \times 3$ patch. The writer receives this $3 \times 3$ observation along with the agent's relative location (with respect to the starting point), updating the memory with this information. Two readers receive randomly chosen $3 \times 3$ and $9 \times 9$ queries, view the current map memory built by the writer, and infer the possible locations of those queries.

to all odd indices in $[1, ((m - n + 2) \cdot 2^{n-1} - 1) \cdot 2 - 1]$ at level $n + 1$. Together, with $2 \cdot i$ and $2 \cdot i - 1$ transferring, the nearest-neighbor up-sampling transfers information from location $1 \le i \le (m - n + 2) \cdot 2^{n-1} - 1$ in level $n$ to all indices in $[1, ((m - n + 2) \cdot 2^{n-1} - 1) \cdot 2]$ at level $n + 1$.

Furthermore, the following convolution operator with $3 \times 3$ kernel size can continue to transfer information from $[1, ((m - n + 2) \cdot 2^{n-1} - 1) \cdot 2]$ to $[1, ((m - n + 2) \cdot 2^{n-1} - 1) \cdot 2 + 1]$ at level $n + 1$. We have $((m - n + 2) \cdot 2^{n-1} - 1) \cdot 2 + 1 = (m + 1 - (n + 1) + 2) \cdot 2^n - 1$. Taking together indices $i$ and $j$, information from location $(i, j)$ where $1 \le i, j \le (m - n + 2) \cdot 2^{n-1} - 1$ in level $n$ can be transferred to $(i', j')$ in level $n + 1$, where $1 \le i', j' \le (m + 1 - (n + 1) + 2) \cdot 2^n - 1$. ■

# B EXPERIMENT DETAILS

## B.1 SPATIAL NAVIGATION: ARCHITECTURES

All experiments related to spatial navigation tasks use multigrid writer-reader(s) architectures. Figure 8 visualizes this architecture and problem setup. At each time step during training, the agent takes a one-step action (*e.g.*, along a spiral trajectory) and observes its $3 \times 3$ surroundings. This observation, together with its location relative to the starting point, are fed into the writer, which must learn to update its memory. The agent has no knowledge of its absolute location in the world map. Two random $3 \times 3$ and $9 \times 9$ patches within the explored map are presented to the agent to as queries (some experiments use only $3 \times 3$ queries). These queries feed into two readers, each viewing the same memory built by the writer; they must infer which previously seen locations match the query.

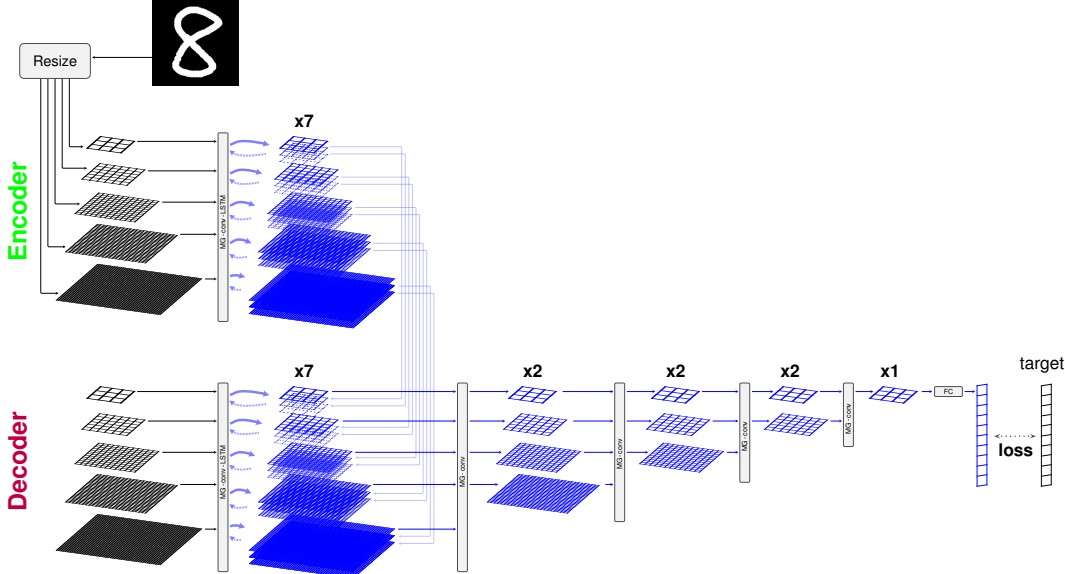

Figure 9: **Multigrid memory encoder-decoder architecture for MNIST sorting.** After processing the input sequence, the encoder (top) transfers memory into the decoder, which predicts the sequence of classes of the input digits in sorted order.

Since the agent has no knowledge of its absolute location in the world map, the agent builds a map relative to its initial position (*map re-centered* in Figure 8) as it navigates.

During training, the writer learns to organize and update memory from localization losses simultaneously backpropagated from the two readers. During inference, only the writer updates the memory at each time step, and the readers simply view (*i.e.*, without modification) the memory to infer the query locations. It is also worth noting that only $3 \times 3$ patches are fed into the writer at each time step; the agent never observes a $9 \times 9$ patch. However, the agent successfully integrates information from the $3 \times 3$ patches into a coherent map memory, in order to correctly answer queries much larger than its observations. Figure 3 (right) shows that this learned memory strikingly resembles the actual world map.

### B.2 SPATIAL NAVIGATION: LOSSES

Given predicted probabilities and the ground-truth location mask (Figure 8), we employ a pixel-wise cross entropy loss as the localization loss. Specifically, letting $S$ be the set of pixels, $p_i$ be the predicted probability at pixel $i$, and $y_i$ be the binary ground-truth at pixel $i$, the pixel-wise cross entropy loss is computed as follows:

$$-\sum_{i \in S} y_i \log(p_i) + (1 - y_i) \log(1 - p_i) \tag{7}$$

### B.3 ALGORITHMIC TASKS: ARCHITECTURES

**Associative Recall Tasks** We employ writer-reader architectures for the associative recall tasks. The architectures are similar to those of the spatial navigation tasks depicted in Figure 8, with some modifications appropriate to the tasks:

- *Standard variant*. In the standard version of the task, we use the same writer architecture shown in Figure 8. For the reader, after progressing to the finest resolution ($48 \times 48$) corresponding to the memory in the writer, the second half of MG-conv layers progressively scale down the output to $3 \times 3$ to match the expected output size (instead of $48 \times 48$ as in the spatial navigation tasks).

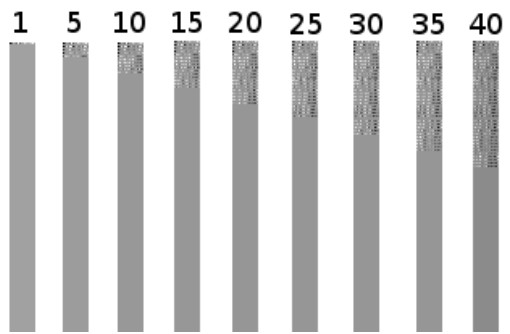

Figure 10: **Visualization of DNC memory in mapping task.** Due to its defined addressing mechanism, the DNC always allocates a new continuous memory slot at each time-step. It does not appear to maintain an interpretable structure of the map.

- *MNIST recall + classification.* For the MNIST variant, since the input is $28 \times 28$ images, we resize input images to 5 scales from $3 \times 3$ to $48 \times 48$ and maintain the same 5-scale structure for 7 layers of the writer. The writer architecture is the same as the encoder architecture in MNIST priority sort task, as depicted in Figure 9. The reader for the MNIST variant is similar to the reader in the standard variant, with the final layer followed by a fully connected layer to produce a 10-way prediction vector over MNIST classes.

**Priority Sort Tasks**   Our priority sort experiments utilize encoder-decoder architectures.

- *Standard variant.* The encoder has the same architecture as the writer used in the associative recall and spatial navigation tasks (Figure 8). For the decoder, the first half of the layers (MG-conv-LSTM) resemble the encoder, while the second half employ MG-conv layers to progressively scale down the output to $3 \times 3$.
- *MNIST sort + classification.*   Figure 9 depicts the encoder-decoder architecture for the MNIST variant.

### B.4    Algorithmic Tasks: Losses

**Standard variants**   We use pixel-wise cross-entropy loss for the standard variants, as described in Section B.2.

**MNIST variants**   For MNIST variants, we use cross-entropy loss over a softmax prediction of the classes. Specifically, letting $C$ be the set of available classes, $p_c$ the softmax output for class $c$, and $y$ a one-hot vector of the ground-truth label, we compute the loss as:

$$-\sum_{c \in C} y_c \log(p_c) \tag{8}$$

### C    DNC Details

We use the official DNC implementation (https://github.com/deepmind/dnc), with 5 controller heads (4 read and 1 write), a memory vector length of 16 elements, and 500 memory slots (8K total), which is the largest memory size permitted by GPU resource limitations. Controllers are LSTMs, with hidden state sizes chosen to make total parameters comparable to other models in Table 1 and Table 2. DNC imposes a relatively small cap on the addressable memory due to the quadratic cost of the temporal linkage matrix (https://github.com/deepmind/dnc/blob/master/dnc/addressing.py#L163).

A visualization of DNC memory in the spatial mapping task ($15 \times 15$ map) is provided in Figure 10.

### D    Runtime

On spatial mapping (with $15 \times 15$ world map), the runtimes for one-step inference with the Multigrid Memory architecture ($0.12\,\text{M}$ parameters and $8\,\text{K}$ memory) and DNC ($0.75\,\text{M}$ parameters and $8\,\text{K}$ memory) are (mean $\pm$ std): $0.018 \pm 0.003$ sec and $0.017 \pm 0.001$ sec, respectively. These statistics are computed over 10 runs on a NVIDIA Geforce GTX Titan X.

## E    DEMOS

- Instructions for interpreting the video demos:
  https://drive.google.com/file/d/18gvQRhNaEbdiV8oNKOsuUXpF75FEHmgG
- Mapping & localization in spiral trajectory, with $3 \times 3$ queries:
  https://drive.google.com/file/d/1VGPGHqcNXBRdopMx11_wy9XoJS7REXbd
- Mapping & localization in spiral trajectory, with $3 \times 3$ and $9 \times 9$ queries:
  https://drive.google.com/file/d/18lEba0AzpLdAqHhe13Ah3fL2b4YEyAmF
- Mapping & localization in random trajectory:
  https://drive.google.com/file/d/19IX93ppGeQ56CqpgvN5MJ2pCl46FjgkO
- Joint exploration, mapping & localization:
  https://drive.google.com/file/d/1UdTmxUedRfC-E6b-Kz-1ZqDRnzXV4PMM

