# OpenReview forum: "Multigrid Neural Memory"
_ICLR.cc/2020/Conference — Reject_

### Official Review · AnonReviewer3 · 2019-10-22
**Official Blind Review #3**

**Rating:** 3

**Review:**

This paper proposes the multigrid memory networks which combine multigrid convolutional layers with LSTMs and evaluates its performance on a reinforcement learning-based navigation task and two algorithmic tasks of priority sorting and associative recall.

The authors claim that by integrating LSTM within the layers of the network, it affords larger memory size while remaining parameter efficient compared to other memory-augmented networks like the DNC, which abstract its memory module as a separate unit from its computational units.

The authors show with their experiments that multigrid memory networks outperform DNC and other models that lack its multigrid inference property.


While the experiments show the proposed networks’ superior performance over baselines, my main concern about this paper is the lack of comparison with more recent memory-augmented models [1,2].

Moreover, in all the experiments, the DNC baseline has memory sizes that are much smaller than the multigrid memory networks. To my understanding, the memory module of DNC can be scaled up without increasing the number of computational parameters. Why is the DNC’s memory not scaled to match that of the multigrid memory networks? It would seem unfair to compare against a baseline with much smaller memory in memory-intensive tasks.


Other comments:


1) How are the input tensors upsampled in the multigrid memory layers?

2) How is the visualization on the right of Figure 3 generated? It
would be more convincing to compare it with that of DNC.

3) Clarity of some parts, especially Section 3.1 & 3.2, could be
improved with more formal mathematical statements about the multigrid
memory networks.

4) How would multigrid memory networks generalize to other tasks that
do not involve input images?



[1] Emilio Parisotto and Ruslan Salakhutdinov. Neural map: Structured
memory for deep reinforcement learning. ICLR, 2018.


[2] Arbaaz Khan, Clark Zhang, Nikolay Atanasov, Konstantinos Karydis,
Vijay Kumar, Daniel D. Lee. Memory augmented control networks. ICLR,
2018.

**Experience Assessment:**

I have published one or two papers in this area.

**Review Assessment: Checking Correctness Of Derivations And Theory:**

N/A

**Review Assessment: Checking Correctness Of Experiments:**

I assessed the sensibility of the experiments.

**Review Assessment: Thoroughness In Paper Reading:**

I read the paper at least twice and used my best judgement in assessing the paper.

---

> ### Author Response · Authors · 2019-11-15
> **Response to Reviewer #3**
>
>
> "comparison with more recent memory-augmented models [1,2]"
>
> DNC is the only state-of-the-art memory architecture that has an official code release.  Other methods mentioned by reviewers do not provide source code, which drastically increases the difficulty of comparison.
>
> We were interested to compare against the Neural Map paper [1], which also has 2D structured memory.  During the course of research, we asked the authors of Neural Map for their implementation, but did not receive a response.
>
> A notable difference from [1] is that we force our memory network to solve a more difficult task.  While [1] also explores spatial mapping, it anchors (hand-codes) memory updates to the current spatial location of the agent.  In contrast, our multigrid memory network must learn which memory cells to update.
>
> The memory component from [2] is directly taken from DNC without any additional modification, so DNC is a good proxy for comparison.
>
> ----
>
> "DNC baseline has memory sizes that are much smaller"
>
> This is a fundamental limit of the official DNC implementation, which has a hidden cost of maintaining the Temporal Linkage matrix (https://github.com/deepmind/dnc/blob/master/dnc/addressing.py#L163).  For N memory slots, this matrix either incurs O(N^2) cost in space or requires an approximation.  We used the largest DNC that would fit in GPU memory.
>
> We have now trained a smaller Multigrid Memory model on the spatial mapping task for the 15x15 world map, spiral motion pattern, 3x3 FoV and Query (new result in Table 1).  Comparing with DNC on the same task:
>
> DNC (0.75M Params, 8.00K Memory): [Prec = 91.09, Recall = 87.67, F = 89.35]
> MG  (0.12M Params, 7.99K Memory): [Prec = 99.79, Recall = 99.88, F = 99.83]
>
> The strictly smaller (in both params and memory) multigrid model perfectly masters the task, while the DNC does not.
>
> Here, the DNC's 8K memory is organized as 500 slots of 16 channels, and 500 slots is more than adequate to store a 15x15 world map (225 locations).  To the Appendix, we added a visualization of DNC memory contents for this task.  The learned memory access strategy, rather than memory size, appears to be the stumbling block for the DNC.
>
> We are also training a multigrid model with 8K memory on the 25x25 world map.  As of now, it is partially trained, but already significantly outperforms the DNC.  On the 25x25 map, spiral motion, 3x3 FoV and Query, we have:
>
> DNC (0.68M Params, 8.00K Memory): [Prec = 77.63, Recall = 14.50, F = 24.44]
> MG* (0.17M Params, 7.99K Memory): [Prec = 96.48, Recall = 96.19, F = 96.33]
>
> (*) partially trained
>
> We will add results for this multigrid model, fully-trained, to the final paper version.
>
> ----
>
> "How are the input tensors upsampled in the multigrid memory layers?"
>
> We use nearest-neighbor upsampling.
>
> ----
>
> "How is the visualization on the right of Figure 3 generated?"
>
> We show the mean across channels of the hidden states in the deepest layer, highest resolution grid.  For comparison, we added Figure 10 (in Appendix) to visualize DNC memory.
>
> ----
>
> "Clarity of some parts, especially Section 3.1 & 3.2"
>
> We have added a formal description of multigrid memory layers to Section 3.1.  To the Appendix, we have added detailed architectures used in the experiments, complementing the more abstract descriptions in Section 3.2.
>
> ----
>
> "How would multigrid memory networks generalize to other tasks that do not involve input images?"
>
> [Quoting our reply to Reviewer #1 regarding a similar question]
>
> Our experiments already cover non-visual sequential data.  There is a distinction between the input having visual structure (i.e., structure over which convolutional operations are helpful), and the network having convolutional layers.
>
> Our spatial mapping task, and a version of each of our algorithmic tasks, all utilize 3x3 input item size.  Here, the first network layer is fully connected: a 3x3 conv filter on a 3x3 input is exactly equivalent to a fully connected layer on a 1x9 input vector.  Moreover, for priority sort and associative recall, the 3x3 items are random - they have no internal visual structure.
>
> Priority sort or associative recall of random 9-element vectors has nothing to do with spatial reasoning.  It is actually a testament to generality that Multigrid Memory, which is internally biased to a 2D spatial memory layout, accomplishes these tasks just as well as the DNC.

---

### Official Review · AnonReviewer1 · 2019-10-23
**Official Blind Review #1**

**Rating:** 3

**Review:**

The paper proposes a multigrid memory architecture by introducing multigrid CNN [1] into convolutional LSTM network [2]. The method extends the convolutional LSTM with bigger memory capacity in forms of multigrid CNNs. Some specific designs such as multiple threads and encoder-decoder are also proposed. The model is validated with synthetic tasks such as spatial mapping, associative recall and priority sort.

Pros:
* The method is well-motivated. Utilizing multigrid hierarchy enables convolutional LSTM to operate across scale space and thus may achieve richer representation for the hidden state memory.
* The experiments are well designed (especially RL tasks), demonstrating the advantage of the proposed model.

Cons:
* The proposed model is not presented clearly. The paper does not show details on how [1] and [2] are integrated, which requires the reader to refer back to the old works and make inference on the integration. Besides graphic illustration, the authors should include a brief review on [1, 2] and introduce some basic formulas describing the combination between the two.
* Section 3.2 is supposed to contain the most important design considerations, but details seem missing. For example, what does the Writer do to the memory?
* The contribution is rather incremental. Without detailed description, it seems that the proposed model is a straightforward replacement of the vanilla CNN with another CNN (multigrid CNN) in the convolutional LSTM architecture.
* The experiments are not very satisfactory for the “generality” claim that the paper makes.

Questions and concerns:
* Could you explain the term “addressable memory space”? It seems that your network’s memory comes from the internal states of LSTM. How is the memory addressable?
* The analysis on information routing seems interesting. However, how does it relate to the memorization capacity? Is there any guarantee that the information from source grid is preserved in higher levels/layers? How does it differ from using vanilla multiple-layer neural networks?
* As DNC is not originally designed for image inputs, how did you feed the images to DNC? Did you tune DNC carefully by adjusting the number of elements per memory slot? Also, DNC seems not a really strong baseline. Other solutions to increase memory capacity of MANNs exist [3, 4]
* For algorithmic tasks, why don’t you include ConvLSTM as a baseline? Also, NTM maybe a better baseline than DNC for these tasks.
* What is the model size and computational complexity compared to other MANNs?
* The model is naturally fit for visual inputs. Is there any advantage when applying it to other sequential data (NLP, time-series)?

Reference
[1] Ke, Tsung-Wei, Michael Maire, and Stella X. Yu. "Multigrid neural architectures." In Proceedings of the IEEE Conference on Computer Vision and Pattern Recognition, pp. 6665-6673. 2017.
[2] Xingjian, S. H. I., Zhourong Chen, Hao Wang, Dit-Yan Yeung, Wai-Kin Wong, and Wang-chun Woo. "Convolutional LSTM network: A machine learning approach for precipitation nowcasting." In Advances in neural information processing systems, pp. 802-810. 2015.
[3] Rae, Jack, Jonathan J. Hunt, Ivo Danihelka, Timothy Harley, Andrew W. Senior, Gregory Wayne, Alex Graves, and Timothy Lillicrap. "Scaling memory-augmented neural networks with sparse reads and writes." In Advances in Neural Information Processing Systems, pp. 3621-3629. 2016.
[4] Hung Le, Truyen Tran, and Svetha Venkatesh. Learning to remember more with less memorization. In International Conference on Learning Representations, 2019. URL
https://openreview.net/forum?id=r1xlvi0qYm.


**Experience Assessment:**

I have published in this field for several years.

**Review Assessment: Checking Correctness Of Derivations And Theory:**

I assessed the sensibility of the derivations and theory.

**Review Assessment: Checking Correctness Of Experiments:**

I assessed the sensibility of the experiments.

**Review Assessment: Thoroughness In Paper Reading:**

I read the paper thoroughly.

---

> ### Author Response · Authors · 2019-11-15
> **Response to Reviewer #1 [2/2]**
>
> "experiments are not very satisfactory for the 'generality'" and "visual inputs" vs "sequential data"
>
> On the contrary, our experiments already cover non-visual sequential data.  There is a distinction between the input having visual structure (i.e., structure over which convolutional operations are helpful), and the network having convolutional layers.
>
> Our spatial mapping task, and a version of each of our algorithmic tasks, all utilize 3x3 input item size.  Here, the first network layer is fully connected: a 3x3 conv filter on a 3x3 input is exactly equivalent to a fully connected layer on a 1x9 input vector.  Moreover, for priority sort and associative recall, the 3x3 items are random - they have no internal visual structure.
>
> Priority sort or associative recall of random 9-element vectors has nothing to do with spatial reasoning.  It is actually a testament to generality that Multigrid Memory, which is internally biased to a 2D spatial memory layout, accomplishes these tasks just as well as the DNC.
>
> Finally, on the algorithmic tasks that operate on 28x28 image input and mix in classification, we are more general than the DNC (which does poorly here).  Multigrid Memory can learn to behave as a flexible combination of CNN-like and DNC-like abilities, as demanded by the task.
>
> We agree that applying Multigrid Memory to NLP tasks is a promising avenue for future work.  Our experiments already cover two domains: reinforcement learning with spatial reasoning (Table 1) and algorithmic tasks (Table 2), comparable in breadth to the experiments introducing NTMs.  Additionally, Multigrid CNNs are a special case (a strict subnet) of our design; Ke et al. already demonstrated their superior performance on vision tasks, including ImageNet classification.
>
> ----
>
> On: "information from source grid" being "preserved in higher levels/layers"
>
> What is stored on each grid is entirely learned.  There is no constraint that any grid duplicate or preserve the information on another.  Upsampling/downsampling of grids occurs when preparing the input to a subsequent convolutional LSTM layer, acting as mechanism for information flow across pyramid scales.
>
> ----
>
> "how did you feed the images to DNC?"
>
> We feed all data to the DNC as vectors, reshaping if necessary.  As explained above, most of our experimental tasks involve 3x3 inputs, which should not be regarded as "images"; initial layers of both Multigrid Memory and DNC are fully-connected in those cases.
>
> ----
>
> "other solutions to increase memory capacity [3,4]"
>
> Source code is not available for these methods, making comparison difficult.  In contrast, we will release code to foster open research.
>
> For fair comparison at equal memory capacity, we trained a smaller Multigrid Memory model; see new results in Table 1 and our response to Reviewer #3.
>
> ----
>
> "ConvLSTM as a baseline?"
>
> For the spatial mapping task, ConvLSTM baselines are provided and perform poorly (Table 1).
>
> We are now training ConvLSTM baselines on the algorithmic tasks, and will include those results in the final version of the paper.  Training is only partially complete, at 400,000 iterations, as of this response.  These partially trained baseline ConvLSTMs have error rates much higher than those for multigrid or DNC (Table 2):
>
> ConvLSTM-deep error rates:
> Associative Recall: [Standard variant: 0.500896,  MNIST variant: 0.887097]
> Sorting: [Standard variant: 0.381804, MNIST variant: 0.565312]
>
> ConvLSTM-thick error rates:
> Associative Recall: [Standard variant: 0.108871, MNIST variant: 0.891129]
> Sorting: [Standard variant: 0.502153, MNIST variant: 0.887500]
>
> ----
>
> "NTM maybe a better baseline than DNC"
>
> DNC is the improved version of NTM with better addressing and memory allocation.  Furthermore, DNC has an official code release (which we use), while NTM does not.
>
> ----
>
> "model size and computational complexity"
>
> Tables list parameter counts and memory capacity.  We have added several Appendix sections, providing more architectural details and wall-clock inference runtime.

---

> ### Author Response · Authors · 2019-11-15
> **Response to Reviewer #1 [1/2]**
>
>
> "details on how [1] and [2] are integrated", "formulas describing the combination"
>
> We revised Section 3.1 to include formulas specifying the exact functionality of our multigrid memory layer.  This presentation now fully details the integration of [1] and [2], with equations elaborating the diagram in Figure 1.
>
> ----
>
> "Section 3.2", "what does the Writer do to the memory?"
>
> Subnetworks are distinguished as "reader" and "writer" solely based on whether they contain parameters that govern the update of LSTM state.  The "writer" subnet is a Multigrid convolutional-LSTM; the "reader" subnet is a Multigrid CNN.  The "writer" subnet contains memory distributed alongside computation, whereas the "reader" subnets contain only computation.
>
> There is feedforward communication from the writer to the readers: as an additional input, the convolution layers in the readers receive the current hidden state of the corresponding LSTMs in the writer.  But, readers cannot effect changes to that state.
>
> ----
>
> On: "overall contribution", "addressable memory space", and "information routing"
>
> Our contribution is to obtain emergent behavior (a memory subsystem) by training a network consisting of basic building blocks connected via a superior wiring pattern (multigrid connectivity).  We achieve with merely multigrid wiring all that the DNC does with extensive custom-designed components (controllers, addressing modes).  To summarize:
>
> (1) A multigrid wiring architecture endows a neural network with exponentially more efficient internal information routing pathways, compared to standard designs.  With multigrid wiring, a few successive layers of a convolutional network can approximate the connectivity of a fully-connected network.
>
> (2) Network parameters determine how to utilize these pathways in a dynamic input-dependent manner.  Ke et al. demonstrate such networks (Multigrid CNNs) can learn to accomplish tasks requiring attentional behavior.
>
> (3) Distributing memory units (LSTMs) throughout the network, the intrinsic capacity to implement attention translates into a capacity to attend to specific memory locations.  Our experiments demonstrate that Multigrid convolutional-LSTMs learn to behave like a large, addressable memory space.
>
> (4) This behavior emerges as a result of training for tasks that benefit from having memory.  Trained multigrid memory networks modify their memory in a localized and context-appropriate manner, directing reads and writes to specific memory cells.  Visualizations clearly show that our maze-exploring agent writes local observations into a coherent map of the environment contained within its memory.
>
> For more detail on these points, please see our overview response above.
>
> ----
>
> "model is a straightforward replacement of the vanilla CNN with another CNN (multigrid CNN) in the convolutional LSTM architecture"
>
> More accurately, we replace convolutional layers in the multigrid architecture with convolutional LSTM layers.  Multigrid wiring weaves together many distinct convolutional LSTMs (one for each pyramid level in each layer), so that their collective behavior can emulate a large-scale memory store.
>
> This straightforward strategy has huge practical consequences: we achieve all the capabilities of the DNC (and more*) from a network composed of basic components.  A powerful, yet simple, design is a virtue.
>
> (*) As the DNC fails to master the spatial navigation tasks, multigrid memory is strictly more capable.

---

### Official Review · AnonReviewer2 · 2019-10-23
**Official Blind Review #2**

**Rating:** 6

**Review:**

Recurrent neural networks that can grow their memory capacity independent of the number of training parameters are an interesting topic. DNC, memory networks and NTM (all cited in this work) are some examples.

This work proposes an architecture inspired by an approach used in the computer vision literature, a multi-scale CNN. However, each cell of the CNN here is a convolutional LSTM.

This approach allows the memory capacity of the architecture to be increased (by increasing the number of cells) while maintaining a fixed number of parameters. The multi-scale nature of it allows memory operations across multiple scales the 2D grid in an efficient manner.

They test this architecture on a mapping and localization task (a natural fit for the multi-scale architecture) and find it outperforms other architectures including a single scale version of the same architecture.

They also compare against the DNC on tasks similar to that used in the original paper (priority sort) and associative recall and again find it learns in fewer iterations and achieves good performance.

Overall, this architecture, while not groundbreaking, is novel in this context and the results show empirical gains. The paper is fairly well written.

This work could be improved by providing more detail (e.g. in the appendix) on the losses and approach used in the navigation task (the only explicit discussion of the loss used in the navigation task is in figure 3). It would also be helpful to provide more detail on the other tasks in the appendix.

Finally, there is little analysis (either theoretical or empirical) on the runtime and memory requirements of this model. For example, figure 6 would seem to imply this model is running slower than the DNC (already quite a slow model) since it has completed less iterations? At a minimum, some empirical numbers of run time speed and memory usage compared with the DNC would be helpful.

**Experience Assessment:**

I have published one or two papers in this area.

**Review Assessment: Checking Correctness Of Derivations And Theory:**

I assessed the sensibility of the derivations and theory.

**Review Assessment: Checking Correctness Of Experiments:**

I assessed the sensibility of the experiments.

**Review Assessment: Thoroughness In Paper Reading:**

I read the paper at least twice and used my best judgement in assessing the paper.

---

> ### Author Response · Authors · 2019-11-15
> **Response to Reviewer #2**
>
> "work could be improved by providing more detail"
>
> We have added a new section (Experiment Details) to the Appendix.  This section diagrams the precise reader-writer architecture used in the navigation (spatial mapping) task, as well as the precise encoder-decoder architecture used in the MNIST sorting task.  Included are details on all network inputs, outputs, and losses applied during training.
>
> ----
>
> "the only explicit discussion of the loss used in the navigation task is in figure 3"
>
> Section 4.1 explicitly stated the form of the loss: "We used a pixel-wise cross-entropy loss over predicted and true locations."
>
> Please also see our answer to the previous question and the new section (Experiment Details) we have added to the Appendix.
>
> ----
>
> "runtime and memory requirements"
>
> Figure 6 concerns model training, not inference time.  The x-axis is training iterations, while the y-axis is loss.  These plots show that for all three tasks (spatial mapping, priority sort, associative recall), the multigrid memory model learns faster (reaches lower loss in fewer steps) than competing models.  For associative recall (rightmost plot), we stopped training the multigrid model at 500,000 steps because it already achieved low loss; in contrast, the DNC, trained for longer (2,000,000 steps) still has much higher loss.
>
> For wall-clock inference time, we have added an Appendix section comparing the runtime of a single forward pass through Multigrid Memory and DNC.  Here, both models have 8K memory cells (a configuration added to Table 1 in response to another reviewer request).  Multigrid Memory takes 18ms (+- 3ms) for inference, whereas DNC takes 17ms (+- 1ms), averaged over 10 runs.
>
> Overall, Multigrid Memory is CNN-like in resource usage.  To roughly estimate the resource requirements of Multigrid Memory, one can take a Multigrid CNN with equivalent depth, grid scales, and channel counts, and multiply GPU compute and GPU memory usage by a constant overhead factor to account for LSTM cell and hidden states.

---

### Author Response · Authors · 2019-11-15
**Response to Reviews and Updates to Paper**

We thank the reviewers and address comments in individual replies.  Acting on reviewer suggestions, we have updated the paper with more technical details (Section 3.1 and Appendix) and new results.  Notably, at equal memory capacity (8K), Multigrid Mem+CNN still significantly outperforms the DNC on the spatial mapping task (Table 1).

We also want to clarify our view of the motivations, contributions, and (high-level) technical insight of the work; this perspective follows.

----

Multigrid memory is a radical new approach to endowing neural networks with access to long-term memory.  It is a drastic simplification compared to all prior memory designs: instead of custom controllers, addressing modes, and/or external storage cells, we have wires.  We take well-known basic building blocks (convolution, LSTMs), wrap them in a multigrid wiring pattern, and achieve all the capabilities of far more complex designs.  Replacing the complexity of the DNC with a wiring pattern is exactly the kind of simplification that advances science.  The results we present here are cause for drastically rethinking the prevailing strategies for designing neural network architectures.

Our networks, once trained, behave like memory subsystems - this is an emergent phenomenon.  Reviews ask about "addressable memory space"; our design contains no explicit address calculation unit, no controller, no attention mask computation.  Yet, our trained networks modify their memory in a localized and context-appropriate manner, directing reads and writes to specific memory cells.  Our maze-exploring agent writes local observations into a coherent map of the environment contained within its memory.  This is possible because learned parameters govern how information flows through its multigrid network, what (if any) memory cells to update, and how to update them.

Multigrid wiring provides an exponentially more efficient information routing topology.  Individual layers (convolutional LSTMs), connected within this topology, learn to coordinate in a manner that allows the system as a whole to behave as a useful memory store.  Multigrid topology provides short neuron-to-neuron communication pathways, which are absent in traditional architectures.

Specifically, bi-directional connections (both coarse-to-fine and fine-to-coarse) between pyramids in subsequent layers allow a signal to hop up pyramid levels and back down again (and vice-versa).  As a consequence, pathways connect any neuron in a given layer with every neuron located only O(log(S)) layers deeper, where S is the spatial extent of the highest-resolution grid (that grid has width and height equal to S).  In a traditional network, this takes O(S) layers to occur.  Equivalently stated, using fixed-size convolutional filters, receptive fields grow exponentially faster in multigrid networks than in standard networks.

Quickly (in few layers of depth) gathering, scattering, broadcasting, or re-directing data across the spatial axes of activation tensors, is thus possible with appropriately chosen network parameters.  Notably, attentional behavior can be implemented in terms of such operations and [Ke et al., Section 4.3] show multigrid CNNs actually learn tasks requiring attention over images.

Sprinkling memory cells throughout the network, the capacity for attention translates to a capacity for specifying which memory cells to read and write.  Our results demonstrate this too can be learned.  A simple wiring design thus enables us to build an entire neural memory subsystem out of basic components.  Simplicity is a virtue; our approach stands in stark contrast to those, like the DNC, that try to craft "neural" versions of conventional CPU components.  Thinking carefully about a core property - efficient communication pathways - obviates the need for such complicated heavy-handed designs.

Recently, one wiring design change has had enormous impact: residual connections [He et al., 2016] provide shortcut pathways across depth, facilitating gradient propagation and allowing very deep networks to be trained.  Multigrid wiring is complementary, improving connectivity across an orthogonal aspect of the network: the spatial dimension.  Its impact is also groundbreaking: multigrid wiring exponentially improves internal data routing efficiency, allowing complex behaviors (attention) and subsystems (memory stores) to emerge from training simple components.

---

### Author Response · Authors · 2020-02-23
**Review and Decision Process**

Like other authors, we devoted a significant amount of time and effort during the feedback phase to address the reviews--running additional experiments as requested by the reviewers and drafting a thorough response. However, neither the reviews nor the meta-review suggest that our response was taken into account as part of the decision-making process. No changes were made to the official reviews, while the meta-review does not acknowledge any of the clarifications that we offered in our response. This is inconsistent with the ICLR 2020 meta-review guidelines (https://iclr.cc/Conferences/2020/MetareviewGuide), which state that the AC is expected to "clearly and thoroughly convey this recommendation and reasoning behind it to the authors" with a justification that reflects discussion among the AC and reviewers. Shortly after meta-reviews were released, we raised our concerns with the program chairs. We explicitly stated that we were not asking the PCs to reconsider the decision, but were raising awareness about the quality of the review process. We have yet to receive a response.

Regards,
Matthew Walter, Michael Maire, and Tri Huynh

---

### Decision · Program_Chairs · 2019-12-19

**Decision:**

Reject

**Comment:**

This paper investigates convolutional LSTMs with a multi-grid structure. This idea in itself has very little innovation and the experimental results are not entirely convincing.